# Proteolyzed Variant of IgG with Free C-Terminal Lysine as a Biomarker of Prostate Cancer

**DOI:** 10.3390/biology10080817

**Published:** 2021-08-23

**Authors:** Anna Lokshin, Lyudmila M. Mikhaleva, Eugene I. Goufman, Marina N. Boltovskaya, Natalia B. Tikhonova, Irina I. Stepanova, Alexandr A. Stepanov, Natalia V. Potoldykova, Andrey Z. Vinarov, Paul Stemmer, Vasily Iakovlev

**Affiliations:** 1Departments of Pathology, Medicine, and Obstetrics and Gynecology, Hillman Cancer Center, University of Pittsburgh, Pittsburgh, PA 15213, USA; 2Research Institute of Human Morphology, 117418 Moscow, Russia; mikhalevalm@yandex.ru (L.M.M.); eugene_goufman@mail.ru (E.I.G.); maribolt@mail.ru (M.N.B.); reprod_pathol@morfolhum.ru (N.B.T.); i-ste@yandex.ru (I.I.S.); 9163407056@mail.ru (A.A.S.); 3Institute of Urology and Reproductive Health, Sechenov University, 119048 Moscow, Russia; potoldykova_n_v@staff.sechenov.ru (N.V.P.); avinarov@mail.ru (A.Z.V.); 4Institute of Environmental Health Sciences, Wayne State University, Detroit, MI 48202, USA; pmstemmer@wayne.edu; 5Angiogen LLC, 101000 Moscow, Russia; angiogen@bk.ru

**Keywords:** prostate cancer, IgG proteolysis, diagnostic, cancer biomarkers, plasminogen

## Abstract

**Simple Summary:**

We have discovered that immunoglobulins digested with plasmin, one of the enzymes of blood clotting cascade acquire a capability to bind to one of the chains of plasminogen. We investigate here the mechanisms and localization of such binding. We also show that levels of this digested immunoglobulin molecule are higher in patients with prostate cancer. Therefore, this digested immunoglobulin could serve as a biomarker for the detection of patients with prostate cancer from patients with benign prostate hyperplasia. We observed that the diagnostic accuracy of blood levels of digested immunoglobulins is dramatically higher than that of PSA.

**Abstract:**

The differential diagnosis of prostate cancer is problematic due to the lack of markers with high diagnostic accuracy. We previously demonstrated the increased binding of IgG to human plasminogen (PLG) in plasma of patients with prostate cancer (PC) compared to healthy controls. Heavy and light chains of PLG (PLG-H and PLG-L) were immobilized on 96-well plates and the binding of IgG to PLG-H and PLG-L was analyzed in serum from 30 prostate cancer (PC) patients, 30 patients with benign prostatic hyperplasia (BPH) and 30 healthy controls using enzyme-linked immunosorbent assay (ELISA). Our results demonstrate that IgG from PC sera bind to PLG-H but not to PLG-L. This interaction occurred through the free IgG C-terminal lysine (Lys) that becomes exposed as a result of IgG conformational changes associated with proteolysis. Circulating levels of modified IgG with exposed C-terminal Lys (IgG-Lys) were significantly higher in PC patients than in healthy controls and in BPH. We used Receiver Operating Characteristic (ROC) analysis to calculate the sensitivity (SN) and specificity (SP) of circulating IgG-Lys for differentiating PC from BPH as 77% and 90%, respectively. The area under the curve (AUC) was 0.87. We demonstrated that the diagnostic accuracy of circulating levels of IgG-Lys is much higher than diagnostic accuracy of total PSA (tPSA).

## 1. Introduction

Prostate cancer is the second leading cause of cancer death in the Western world and American men after only lung cancer. According to the American Cancer Society, one in seven men will be diagnosed with PC during their lifetime, and 60% of men over 65 years of age will be diagnosed with the disease [1]. The prediction of prostate cancer in primary care is typically based on serum total prostate-specific antigen (tPSA) and digital rectal examination tests [2]. Unfortunately, these tests are not helpful for differentiating PC from benign conditions, such as benign prostatic hyperplasia (BPH). Patients with BPH can also present with raised levels of tPSA, preventing the accurate stratification of suspected cases of PC and leading to unnecessary invasive biopsies [2,3]. It is, therefore, clear that there is a need for more accurate methods to risk stratifying men who present with symptoms of PC to prevent the over-diagnosis and unnecessary treatment of patients with benign conditions [4]. In the intense search for diagnostic biomarkers for PC, hundreds of potential biomarkers in urine, serum, tissue, and semen proposed have been reported [5,6,7]; however, none of these biomarkers has demonstrated required reliable diagnostic accuracy. Thus, the search for diagnostic biomarkers of PC continues.

Invasion and metastasis, the hallmarks of cancer aggressiveness, are accompanied by the activation of proteolytic enzymes [8]. One such proteolytic enzyme is a serine protease plasmin (PM), a component of the plasminogen (PLG) proteolytic cascade that consists of serine proteases, binding proteins, and inhibitors [9]. The PLG family members have exceptional significance, due to their ability to cleave virtually any element of the extracellular matrix and basement membrane [10]. The components of the PLG activation system are overexpressed in malignant tumors and promote local invasion, metastasis, and angiogenesis [8,11]. PM is mainly derived from inactive PLG by tissue- or urokinase-type plasminogen activators (tPA or uPA) [12]. PM plays an active role in tumor metastasis by digesting tumor adjacent tissues, thus creating space for further malignant growth, creating a pool of nutrients (amino acids) for the growing tumor, and dissolving connective tissue and the basement membrane necessary for invasion and metastasis [8]. Malignant or premalignant prostate lesions have been shown to contain an increase in the deposition of IgG, relative to healthy tissue [13]. The localization of the IgG in the area of increased PLG activity is expected to result in elevated IgG proteolysis in the vicinity of the tumor. PM has been reported to cleave IgG to form an IgG variant with free C-terminal Lys (IgG-Lys) [14,15]. Following PM cleavage, IgG fragments were shown to specifically interact with PLG via their C-terminal Lys [16]. PLG heavy chain (PLG-H) has Lys binding sites, which may bind to IgG-Lys. We have previously demonstrated elevated levels of IgG capable of binding to PLG in plasma of patients with prostate cancer (PC) compared to healthy donors, making such IgG potentially suitable as PC biomarkers [17]. The nature of the IgG-PLG interaction is unclear. Such interaction could occur either through a classical antigen–antibody (Ag-Ab) binding mechanism or via C-terminal Lys mediated binding. Recent studies have reported the potential usefulness of products of proteolytic degradation as cancer biomarkers [16,18,19]. However, as of now, no proteolytic derivatives have been sufficiently validated, so the search continues. Here, we study the mechanisms and localization of IgG/PLG binding in patients with prostate cancer (PC), patients with benign prostate hyperplasia (BPH), and healthy controls, to ascertain the usability of IgG-Lys as a potential biomarker of PC.

## 2. Materials and Methods

### 2.1. Patients

Serum samples from PC (*n* = 30) and BP (*n* = 30) patients were obtained from the urological clinic of Sechenov First Moscow State Medical University, Moscow, Russia. All diagnoses were confirmed using physical examinations, digital rectal examination, and biopsies. Control samples (*n* = 30) were obtained from healthy men (Table 1).

This research was approved by the Ethics Committee at the Sechenov First Moscow State Medical University. Written signed informed consent was obtained from each volunteer before entry into the study. The study conformed to the ethical principles for medical research involving human subjects. All samples were stored at −20 °C.

### 2.2. Plasminogen Preparation

PLG was prepared from the plasma of healthy donors using Lys-sepharose 4B, as described elsewhere, with some modifications [20]. The heavy chain (PLG-H) and light chain (PLG-L) of PLG were purified from PLG after urokinase cleavage of the peptide bond between amino acids 561–562, followed by a reduction in S–S bonds and acetylation. The separation of PLG-H and PLG-L was performed using Lys-Sepharose 4B, as described elsewhere [21].

### 2.3. ELISA

Both PLG-H or PLG-L were immobilized on 96-well ELISA plates (SPL Maxibinding, Korea) as previously described [22]. Each well was filled with 100 μL of 5 µg/mL coating solution of PLG-H or PLG-L in 0.1 M sodium carbonate buffer (pH 9.6), then incubated overnight at 37 °C. Unbound sites were blocked overnight with 1% bovine serum albumin (BSA; MP Biomedicals) in phosphate-buffered saline (PBS) (0.001 M sodium phosphate with 0.1 M NaCl, pH 7.4) at room temperature. The blocking solution was removed, plates were dried at room temperature and stored in vacuum-sealed containers at 4 °C for up to a year. Serum samples were diluted 300-fold with PBS containing 0.5% BSA (BSA-PBS), and 100 μL was added to each well of 96-well plates and incubated for 1 h at 37 °C. The solution was then removed, and plates were washed 4 times with 200 μL PBS containing 0.05% Tween 20. Subsequently, 100 μL of mouse monoclonal anti-human IgG conjugated with horseradish peroxidase (Diateh-Em, Moscow, Russia) were added and incubated for 1 h at 37 °C. Plates were washed with PBS-Tween 20 solution, and 100 μL tetramethylbenzidine solution (Diateh-Em) was added. The reactions were stopped after 25 min using 100 μL 2 M H_2_SO_4_. Optical density at 450 nm (OD) was recorded using a multichannel spectrophotometer (Bio-Rad, Hercules CA USA). Samples were analyzed in duplicate.

PM-treated IgG were prepared by mixing 100 μL PLG (1 mg/mL) in PBS with 100 μL urokinase (5000 IU/mL; Sigma-Aldrich) for 5 min, then adding 100 μL human IgG (1 mg/mL in PBS; Sigma-Aldrich, St Louis MO USA) and incubating for 6 h at 37 °C. The reaction was stopped by adding 50 μL aprotinin (10,000 IU/mL, Sigma Aldrich). Control untreated IgG were prepared by incubation with 100 μL aprotinin instead of urokinase. Both the PM-treated and control IgG were diluted 100-fold by adding BSA-PBS buffer prior to analysis with ELISA.

Lysine competition ELISA was performed with samples diluted with 0.2 M lysine in BSA-PBS buffer and compared with samples diluted in BSA-PBS buffer.

### 2.4. Elimination of C-Terminal Lysine Residues with Carboxypeptidase B (CPB)

100 μL PM-treated IgG (0.1 mg/mL) in PBS or 100 μL sera diluted 100-fold with PBS was mixed with 2 μL CPB (5 mg/mL; Sigma-Aldrich) and incubated for 6 h at 37 °C. The reaction was stopped by adding 2 μL 1,10-phenathroline (Sigma-Aldrich) in 180 mg/mL methanol. Serum samples were diluted 3-fold with BSA-PBS buffer for ELISA.

Immunohistochemistry (IHC) was performed on formalin fixed paraffin embedded sections from samples taken after total prostatectomy. IHC preparations were assessed by a semi-quantitative method in 10 fields of view (FOV). Histological processing was performed using an Excelsior ES apparatus (Thermo Scientific, Pittsburgh, PA, USA). The 4–6 μm sections were stained with hematoxylin and eosin (H&E). Morphometric analysis was performed using a Leica 2500 microscope. Sections were dewaxed, hydrated, the activity of endogenous peroxidase was blocked, antigens were unmasked, the nonspecific binding was blocked, and sections were incubated with primary Abs for 1 h at 37 °C. Primary monoclonal Abs to uPA (#Ab229378, Abcam, Shanghai, China), rabbit polyclonal Abs to ENO1 ((#Ab229378, Abcam), mouse monoclonal Abs to membrane specific antigen (PSMA-5, HyTest, Turku, Finland), mouse monoclonal Abs to human IgG (#MGH IgG, IMTEK, Moscow Russia), mouse monoclonal Abs to Ki-67 (#ab229378, Abcam) were used. For detection, Leica biosystems-Novolink Polymer Detection System was used. Reactions, in which the primary antibodies were replaced with 0.01 M phosphate buffer, served as negative control.

Statistical analysis was performed using AtteStat software (Version 12.0.5). Signal strength from ELISA assays were presented as mean OD values ± SEM. The comparisons of mean OD between cohorts were performed by the Mann–Whitney U-test. Classification power of IgG-Lys and total PSA was evaluated by ROC analysis.

## 3. Results

PLG consists of two chains, heavy (PLG-H) and light (PLG-L). To determine, which chain preferentially binds circulating IgG, we have developed an ELISA assay where we coated 96-well plates with either purified PLG-L or PLG-H. Plates were incubated with serum samples collected from 30 patients with PC, 30 patients with BPH, and 30 healthy controls. To measure binding, plates were incubated with mouse monoclonal HRP-conjugated anti-human IgG Ab. We determined that PLG-H, but not PLG-L coated plates demonstrated IgG binding above the background. OD was significantly higher in PC vs. control or BPH samples (*p* < 0.001 for both comparisons) (Figure 1). BPH did not significantly differ from healthy controls on PLG-H coated plates (Figure 1). On PLG-L coated plates, IgG binding was not significantly different from the background (data not shown), and no differences between PC and two non-cancer controls were observed (Figure 1).

Circulating IgG pool is comprised of full and plasmin (PM) proteolyzed molecules. To further ascertain whether proteolyzed IgG fragments can bind PLG, we used purified human IgG purchased from Sigma Aldrich (cat#I4506). Incubation of purified human IgG with activated PLG resulted in appearance of two additional bands with MW < 45 kDa corresponding to proteolyzed fragments of heavy and light IgG chains (Figure 2).

We next incubated untreated and proteolyzed purified IgG with PLG-H and PLG-L bound to ELISA plates. We observed a significantly stronger binding of proteolyzed vs. untreated IgG to PLG-H (*p* = 0.009) (Figure 3). No binding beyond the background was detected for PLG-L (Figure 3). Thus, the binding pattern for purified IgG closely resembles that for IgG from blood serum of PC patients. Since the PLG-L coated plates did not give significantly different readings in both experiments, for the following experiments we used only the PLG-H coated plates.

Binding of IgG to PLG could occur via classical Ag-Ab binding mechanism where anti-PLG-H IgG Abs specifically recognize and bind PLG-H. It was reported that PM-mediated proteolysis of IgG results in the formation of IgG derivative with free lysine (Lys) moiety on carboxy (C) terminal (IgG-Lys). Since PLG-H has Lys-binding sites, it is possible that an alternative binding mechanism can occur where IgG-Lys binds to these PLG-H lysine binding sites. To determine the mechanisms of IgG/PLG-H binding, we performed competitive ELISA in the presence or absence of 0.2 M Lys to test whether excess Lys will block Lys-binding sites on PLG-H and will consequently prevent the binding of IgG-Lys. We analyzed samples from PC patients and healthy controls. Lys competition resulted in significantly lower binding of circulating IgG to PLG-H in PC samples, but not in samples from healthy controls (*p* < 0.001) (Figure 4A). We then tested the effect of excess Lys in the reaction buffer on binding of PLG-H to purified human IgG, intact or proteolyzed by PM. Lys significantly reduced the binding of PM-treated IgG to PLG-H (*p* = 0.009) (Figure 4B). Binding remained unchanged in samples, where PBS was added instead of 0.2 M Lys.

To further confirm that IgG/PLG-H binding occurs through Lys-binding sites, C-terminal Lys was removed by carboxypeptidase B (CPB) in pooled PC samples (from 10 patients) and in purified human IgG. Treatment with CPB significantly decreased IgG/PLG-H binding in both human samples (*p* < 0.001) and in purified IgG (*p* = 0.009) (Figure 5).

We next analyzed correlations between IgG-Lys levels and markers of prostate malignancy by immunohistochemical (IHC) staining of several markers in FFPE tumor biopsies from 5 patients with high IgG-Lys levels (OD > 1.4) and from 5 patients with low IgG-Lys levels (OD < 0.6). It is well known that protease activity in tumor positively correlates with the degree of malignancy [23]. High protease activity may result in higher circulating levels of cleaved IgG-Lys. Protease activity can be assessed directly by the presence of uPA, a plasminogen activator, or indirectly by expression of plasminogen receptors, such as alpha-enolase (ENOA) on tumor cells, [24,25]. The degree of malignancy was assessed by staining with prostate-specific membrane antigen (PSMA) [26,27]. We additionally used Ki67 staining to evaluate the proliferation of PC cells in tumor biopsies from patients with high and low circulating IgG-Lys levels. Both PC and BPH presented higher expression of all five proteins compared to normal tissue. Staining for uPA, PSMA, and ENO1 was higher in PC than in BPH. The expression of Ki-67 was stronger in BPH, which could be explained by higher cell numbers in BPH than in PC foci. IgG expression varied in both cancer and benign conditions, but in PC it was mostly stained in prostatic ducts (Figure 6). The staining intensity of prostatic duct IgG was similar in both low and high IgG-Lys groups (data not shown).

### 3.1. Blood IgG-Lys in PC, BPH, and Healthy

To decrease mortality from PC, tests are necessary to detect disease earlier and to differentiate malignant disease from BPH. Currently, PSA is the strongest PC biomarker, but its performance requires further improvement [28]. Thus, we performed a preliminary pilot analysis of blood levels of IgG-Lys and total PSA (tPSA) in patients with PC, BPH, and in healthy controls (*n* = 30 in each group). We observed significantly higher circulating levels of PLG-Lys in a subgroup with Gleason score 4 + 3 (*n* = 17) compared to a Gleason score 3 + 4 (*n* = 13) subgroup (Figure 7A). Circulating levels of IgG-Lys were significantly higher in PC than in BPH or healthy controls (Figure 7B), whereas tPSA levels significantly differed only between PC and controls and between BPH and healthy controls but were similar between PC and BPH (Figure 7C).

### 3.2. Receiver Operating Characteristic (ROC) Curve Analysis

We next analyzed potential diagnostic power of circulating IgG-Lys in blood of patients with PC, BPH, and in healthy controls (*n* = 30 in each group). The sensitivity (SN) of the classification of PC from heathy controls with IgG-Lys was 93% at 93% specificity (SP) and the area under the curve (AUC) of 0.98 (Figure 8A). PSA demonstrated stronger classification power for PC vs. healthy with 100% SN at 100% SP with 1.00 AUC (Figure 8A). SN for the classification of healthy individuals from BPH was 93% at 67% SP and AUC of 0.91 (Figure 8B). The classification power of tPSA for PC vs. BPH was much lower than that of IgG-Lys with only 20% SP at 93% SN and an AUC of 0.52 (Figure 8B). Combining IgG-Lys with tPSA for the discrimination of PC from BPH did not improve classification power provided by IgG-Lys alone (data not shown).

## 4. Discussion

In this study, we analyze mechanisms of binding of IgG to PLG and explore the diagnostic potential of circulating modified IgG with a free Lys as a biomarker in patients with PC in comparison with patients with BPH and healthy controls. We demonstrate that circulating IgG treated with PM bind to PLG-H, but not to PLG-L. We also show that this binding occurs via Lys-binding sites on PLG-H. We hypothesize that PM proteolyzes IgG to change its tertiary structure and expose the C-terminal Lys. In unmodified IgG, this C-terminal lysin is sterically hidden and is thus unavailable for Lys-binding sites on PLG-H. We demonstrate high protease activity in PC tissue manifested by intense IHC staining for uPA, PSMA, and ENO1. We hypothesize that low circulating levels of IgG-Lys in BPH could be explained by differences in the structure and permeability of blood vessels between malignant and benign tumors. Hyperpermeable tumor vessels leak plasma IgG at a much higher rate than normal or benign counterparts. Besides in hyperplasic lesions, the undamaged basal membrane could also prevent the penetration of products of proteolysis in circulation.

Currently, PSA is the only biomarker that is approved by the Food and Drug Administration (FDA) for cancer screening [28]. Although PSA is a highly sensitive serum biomarker that has changed the management of prostate cancer, PSA testing is not perfect, since PSA levels increase with age and in other conditions, including BPH and prostatitis [29,30]. Approximately 10% of the male population has a PSA value that is higher than 10 ng/mL but does not have cancer [31]. The US Preventive Services Task Force currently recommends against using it for routine prostate cancer screening (www.uspreventiveservicestaskforce.org/uspstf/recommendation/prostate-cancer-screening, accessed on 20 August 2021). It is, therefore, clear that there is a need for more accurate methods to risk stratifying men who present with symptoms of PC, to prevent the over-diagnosis and unnecessary treatment of patients with benign conditions [32]. Here, we provide pilot data that the classification power of circulating levels of modified IgG with free lysine (IgG-Lys) dramatically exceeds that of tPSA, thus making IgG-Lys a promising candidate biomarker for the differential diagnosis of PC.

Our results require further confirmation in larger sets. Additionally, we need to develop a standardized test for IgG-Lys measurements using standard curve and concentration per milliliter. If successfully confirmed, using IgG-Lys as a biomarker could inform physicians on the necessity of prostate biopsies in males with high PSA and could substantially decrease the number of unnecessary biopsies.

## 5. Conclusions

These data indicate that, in PC, IgG may undergo proteolysis at the site of the tumor resulting in the appearance of IgG-Lys in the circulation of these patients. We provide pilot data that circulating levels of IgG-Lys could serve as a biomarker for the differential diagnosis of PC from BPH.

## Figures and Tables

**Figure 1 biology-10-00817-f001:**
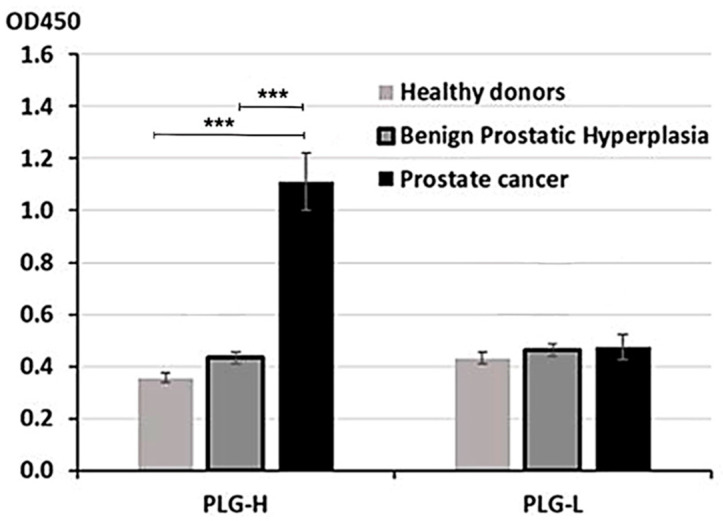
ELISA analysis with PLG-H or PLG-L coated plates. Serum samples from PC, BPH, and healthy donors were incubated in 96-well plates coated with either PLG-H or PLG-L and IgG binding was measured by HRP-conjugated anti-IgG Abs. The experiment was repeated 3 times. *** *p* < 0.0001, not indicated–not significant.

**Figure 2 biology-10-00817-f002:**
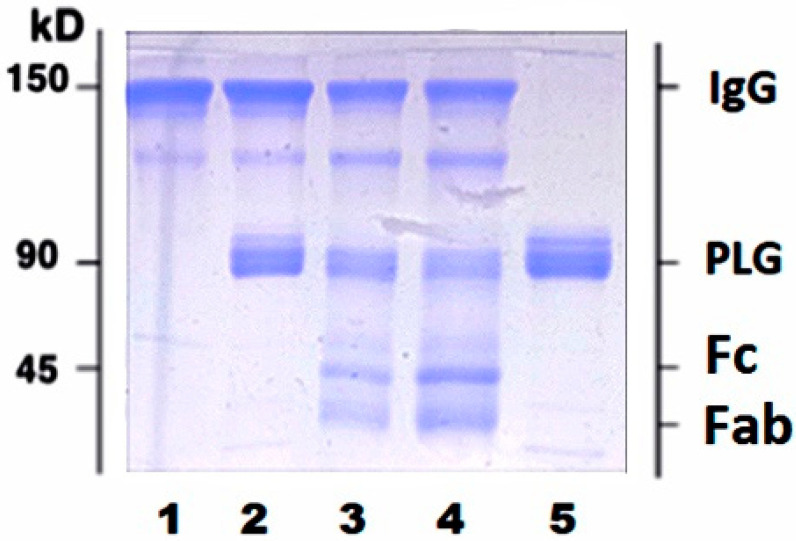
Plasmin-mediated proteolysis of purified human IgG results in appearance of new bands with low MW (<45 kDa). Purified IgG were treated with PM as described in Methods section and analyzed using 10% SDS-PAGE under non-reducing conditions. **1**, untreated IgG; **2**, IgG plus activated PLG without incubation; **3**, IgG plus activated PLG after 1 h incubation; **4**, IgG plus activated PLG after 6 h incubation; **5**, PLG. Bands with MW < 45 kDa in lanes 3 and 4 correspond to Fc and Fab fragments of IgG.

**Figure 3 biology-10-00817-f003:**
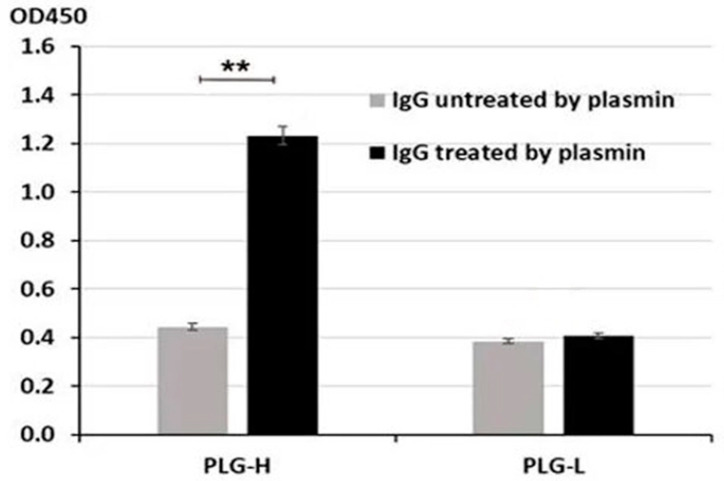
Binding of untreated or PM-proteolyzed purified IgG to PLG-H and PLG-L was evaluated as in Figure 1. The experiment was repeated 3 times. ** *p* < 0.01, not indicated–not significant.

**Figure 4 biology-10-00817-f004:**
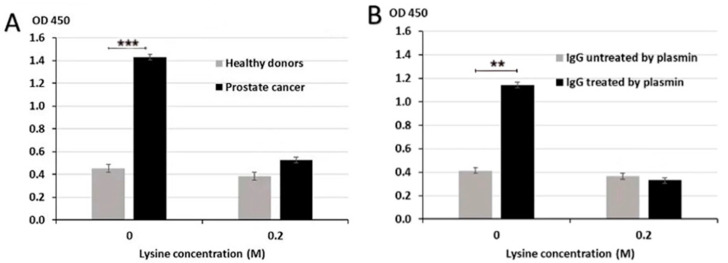
Inhibition of IgG/PLG-H binding by lysine. Lysine at 0.2 M inhibits binding of IgG to PLG-H in PC samples (**A**) and PM treated purified IgG samples (**B**) in a competition ELISA assay. The experiment was repeated 3 times ** *p* < 0.01; *** *p* < 0.001.

**Figure 5 biology-10-00817-f005:**
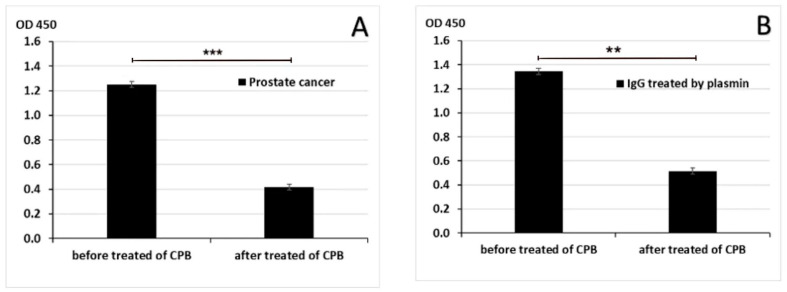
Binding of IgG to PLG-H after removal of C-terminal lysine. CPB inhibited the binding of IgG to PLG-H in PC samples (**A**) and in samples of purified IgG treated with PM (**B**). The experiment was repeated 3 times. ** *p* < 0.01; *** *p* < 0.001.

**Figure 6 biology-10-00817-f006:**
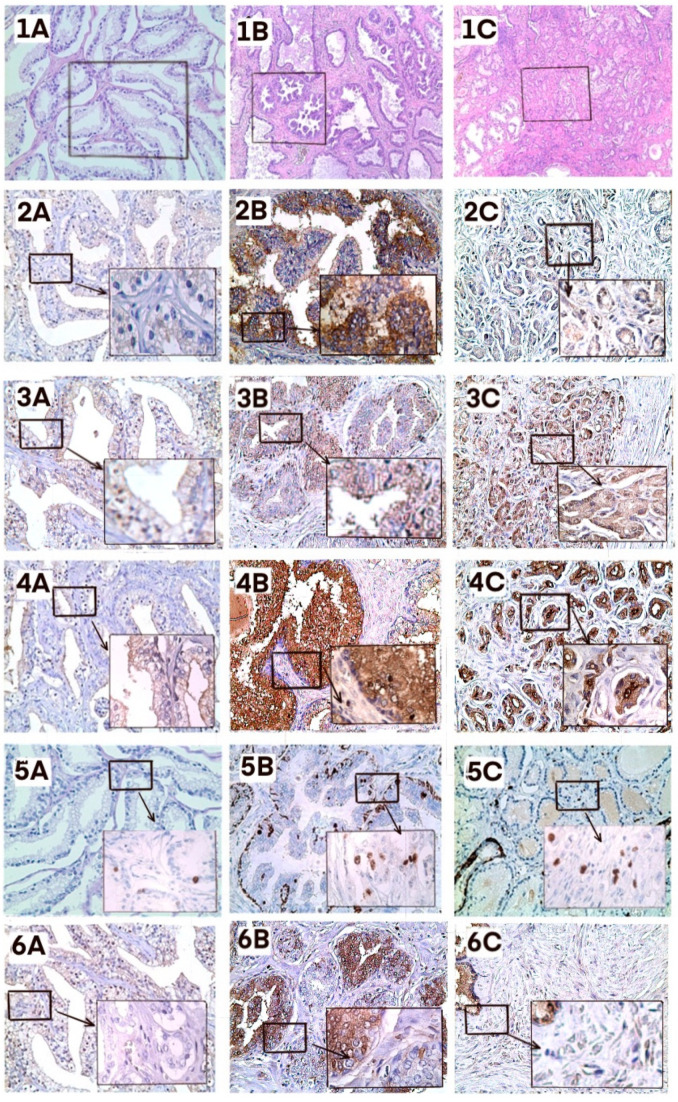
Immunohistochemical staining. Healthy tissue (**A**), BPH (**B**), PC (**C**). **1**, H&E (**1A**, **1B**, **1C** ×50 magnification); **2**, uPA; **3**, ENO1; **4**, PSMA; **5**, Ki-67; **6**, IgG. **A**, **B**, **C**, ×200 magnification; insets, ×400 magnification. Five patients from each group were analyzed. A representative slide is presented.

**Figure 7 biology-10-00817-f007:**
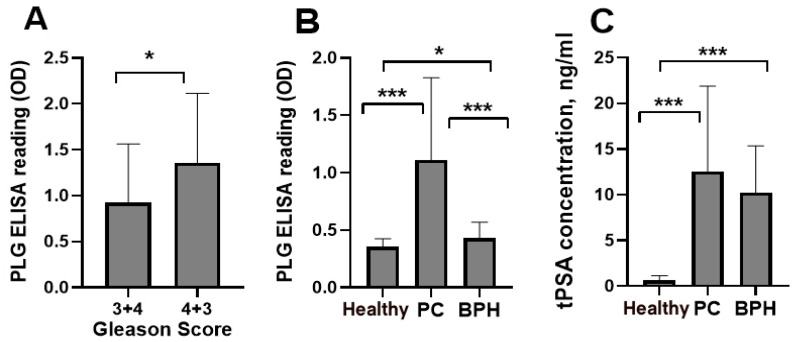
Circulating levels of IgG-Lys measured by PLG ELISA in PC subgroups with Gleason scores (3 + 4) vs. (4 + 3) (**A**) and in PC, BPH, and healthy controls (**B**). Circulating levels of tPSA in PC, BPH, and healthy controls (**C**). * *p* < 0.05; *** *p* < 0.001.

**Figure 8 biology-10-00817-f008:**
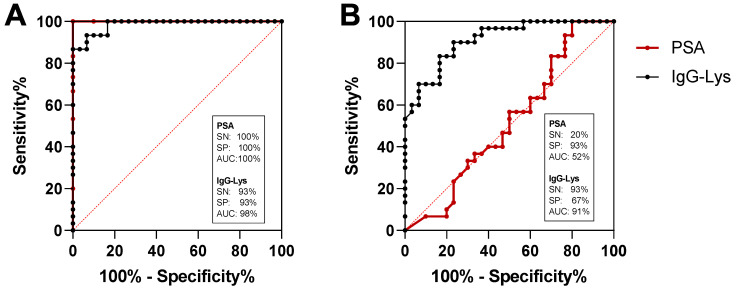
ROC analysis of serum IgG-Lys vs. tPSA in patients with PC vs. healthy donors (**A**), and PC vs. BPH (**B**).

**Table 1 biology-10-00817-t001:** Characteristics of selected populations.

Cohort	Age, Yearsold (Median)	*n*	tPSA, min-max, ng/mL	Staging	
				Gleason Score	Stage
Healthy controls	45–67 (62)	30	0–1.5	-	-
BPH	51–65 (59)	30	6–21	-	-
PC	53–67 (57)	30	6–31.2	3 + 4, *n* = 17 4 + 3, *n* = 13	T1, *n* = 6 T2a, *n* = 17 T3a, *n* = 7

## Data Availability

The data presented in this study are available on request from the corresponding author.

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
