# Peer review of "Proteolyzed Variant of IgG with Free C-Terminal Lysine as a Biomarker of Prostate Cancer"

_biology, 2021, doi:10.3390/biology10080817_

Round 1
Reviewer 1 Report
Where the authors have addressed most of the initial comments they:
- Where inserts of higher magnification of the immunohistochemistry has been included there need to be arrows included to point to the specific staining and a more detailed description of the results in the results section.
- Calibration is a plot of the predicted vs observed risk and should be carried out as there might be a good AUC but the biomarker might over or under predict the correct answer indicating that the biomarker is not well calibrated. See DOI: 10.1016/J.spinee.2021.02.024 for a reference.
Author Response
- In Figure 6, we have added arrows pointing to the origin of high magnification insets. Results section was revised to include a more detailed description of the results.
- A calibration plot shows that the fraction of events matched the predicted probability for subjects with the same scores. As such, calibration analysis (plot) method is not applicable to our current results, as the focus of this paper is on one biomarker and it’s discrimination ability and accuracy. We do not develop a multiparametric decision model, nor we evaluate a validation set, but only present a single cut-off based classification model in the training set. Since calibration is a process we plan to perform it as part of the biomarker validation in our future work.
Reviewer 2 Report
The manuscript entitled “Proteolyzed variant of IgG with free C-terminal lysine as a biomarker of prostate cancer” from Lokshin et al. is of interest since it approaches a known problem using the PSA levels as screening tool. Nevertheless, it lacks consistency and is not clear in some parts. It needs to be rewritten in some parts and the figures/figure legends also need work.
Further the manuscript needs proofreading for typos, missing space between words throughout the manuscript e.g. Prostatecancer in Table 1, proteolysiat line 285…
For numbers and units use the SI rules or the publishers recommended style, but be consistent within the manuscript (e.g. unbreakable space between number and unit with some accepted exception e.g. %). The SI rules and Chicago manual of style might help.
Lane 179-180… and 179+194: Lys-IgG vs IgG-Lys vs IgG-Lys, lysine vs Lysine. Be consistent throughout the manuscript.
Keep an order on presenting results. If the figure shows result A and then result B, also describe in the text result A before result B. This makes the manuscript more readable.
Be consistent with prostate cancer abbreviation. PC vs PCa.
M&Ms:
For reproducibility, the catalogue number of reagents, especially antibodies (companies mostly have more than one antibody in their portfolio) should be added. E.g. in line 157.
Lines 82 and 86: The references 13 and 14 do not support/not related to the claim.
Line 88: Ref 15: please add the chapter or page numbers where to find the protocol for this ELISA.
Be constant with space between numbers and units and magnifications x100 vs x 100. Please follow
Line 112-118: This paragraph in material and methods needs to be re-written. It is confusing and not clear.
Line 121: Field of view (FOV) is more descriptive than fields.
Line 126-128: I assume that only one Ab was used for each protein to detect, not multiple Abs as written.
Line 130: What phosphate buffer was used and was the pH adjusted?
Results:
Line 146: Is IgG or IgG-Lys or both detected?
Line 156: Should read proteolyzed IgG fragments?
Lane 220: staining intensity of what, all above mentioned stainings?
Lane 221: Why is this sentence not within the paragraph 228-236 and it does not match the figure labelling.
Lane 276: The sentence is not clear and there is no table 8.
The preparation/styles of figures and figure legends within a manuscript should be constant. Put the label eg. (A) before the text or after, not like Fig. 5 vs Fig. 6, not conclusive in Fig. 7 at all.
Figure 1, 3, 4: Significant differences are indicated by stars and not significant values are not indicated. Please remove the NS indication since it does not add any value . Please use the same sizes for stars in same/similar figures e.g. figure 4.
Fig 2: Lane 165: please put Fc before Fab since you read the gel from top to bottom, for consistency.
Figure 6: update the figure legend. 1B and 1C inserts are not at a different magnification. 2-6 have no magnification indicated.
Figure 7: The figure needs work since it does not match the text or the figure legend in some parts. Please be consistent, the y-axis labelling is not clear (OD) vs Optical density. The stars for confidence interval in the legend do not appear in the figure. Please indicate the significance by stars as in other figures. IgG-Lys vs PLG-Lys?. Left-right, where is the middle described. The figure legend is not clear.
Please replace the word “strong” in line 287 with “reliable” if applicable or omit it, unless further studies are conducted and the test is suitable for routine use.
Author Response
- The manuscript was proofread, and all typos and missing spaces between words were corrected.
- We have inserted spaces between numbers and units according to SI rules
- Inconsistencies with lysine vs. Lysine, Lys-IgG and IgG-Lys, PC and PCa have been corrected
- The order of presenting results has been corrected
- Catalog numbers for detection Abs have been added in Methods
- References 13 and 14 have been replaced with appropriate references supporting the claims
- Ref 15 was replaced with the one with detailed protocol for ELISA
- The paragraph on lines 112-118 in materials and methods have been re-written for clarity. We emphasized that only one Ab was used to detect each protein
- Composition of phosphate buffer including pH on line 130 has been added
- Is IgG or IgG-Lys or both detected? Only IgG-Lys, but not intact IgG, binds to PLG-H.
- Line 156 has been corrected to read ‘proteolyzed IgG fragments’
- We clarified ‘staining intensity of prostatic duct IgG’ on lane 220
- The sentence from line 221 was moved to the next paragraph as suggested by the reviewer.
- The sentence on lane 276 has been revised for clarity and correctness.
- The styles of figures and figure legends have been revised for uniformity
- In Fig 2, Fc was stated before Fab
- Figure 6 has been substantially revised in response to reviewer’s critique. Magnifications’ descriptions have been corrected for uniformity
- Figure 7 has been revised for clarity and uniformity as suggested by the reviewer:
- The word “strong” in line 287 has been removed.
Round 2
Reviewer 2 Report
The manuscript is significantly improved, but still needs proofreading.
Figure 6 has improved but the insert does not seem to match the indicated location. Please comment on that.
Figure 7: Please change Norm to healthy or include in the figure legend Norm in brackets after healthy controls.
Figure 8: Please write out ROC and also SN, SP, and AUC the first time they appear.
Line 86: remove the bracket
107: Spaces
109: buffered
118: use non-breaking space
119: space
150: space
166: not shown? PLG-L coated plates are shown in figure 1.
171: space
172: Ab vs. Abs. be consistent within the manuscript.
Figure 4A: pasted twice?
230: remove komma
234: komma before and in enumeration, be consistent
281: space
299: space
301: hyperpermeable (space)
309-310: time? has vs. have, do not vs. does not
Author Response
We have addressed all reviewer's comments regarding spelling, spacing, and punctuation.
We have additionally addressed other points by the reviewer:
We have revised Figure 6 replacing the inserts with inserts matched the indicated locations.
We have changed Norm to healthy in Figure 7.
We have spelled out the abbreviations of ROC, AUC, SN, and SP in the Abstract and also in the text on lines 245, 247, 248
166: not shown? PLG-L coated plates are shown in figure 1, but the background values are not.
It does not seem that Figure 4A is pasted twice.
We hope we have addressed all reviewer's comments
This manuscript is a resubmission of an earlier submission. The following is a list of the peer review reports and author responses from that submission.
Round 1
Reviewer 1 Report
In the present paper, Authors analyzed mechanisms and localization of IgG/PLG binding in patients with prostate cancer, benign prostate hyperplasia and healthy controls to evaluate the potential role of IgG-Lys as potential biomarkers of prostate cancer.
Authors conclude that levels of IgG-Lys could complement PSA for differential diagnosis between prostate cancer and benign prostatic hyperplasia achieving 83% sensitivity and 83% specificity.
The efforts of the authors are praiseworthy, the manuscript is well written, references are accurate. However, the paper shows several meaningful methodological limitations.
The main limitation concerns the limited sample size, which may have ultimately undermined reliability of ROC curve and sensibility/specificity values.
Baseline features of cohort study should have been more extensively reported. No data is provided according to tumor histology. In addition, ROC curve was plotted using a PSA cut off of 25.4 ng/ml, although no linear correlation exists between PSA levels and prostate cancer detection.
Extensive English language revision is needed. Along the discussion there are several sentences referring to personal comments and/or written in foreign language.
Reviewer 2 Report
The paper explores the binding of IgG to human plasminogen in plasma of patients with prostate cancer compared to healthy controls. Where the biochemistry and mechanisms aspect of the paper are interesting the diagnostic potential is too preliminary for publication at this time and needs to be supported by a larger cohort of patients and an independent validation cohort and written in line with the TRIPOD guidelines.
In addition
- For the immunohistochemistry the images need to be of higher magnification to show the location of staining.
- The preformance of the biomarker is compared to PSA alone but this is not standard practice and should be compared to current risk calculators such as the PCPT and ERSPC which are used clinically to see if it improves on current clinical practice.
- In addition to ROC and AUC values decision curve and calibration analysis needs to be carried out.